# Genome-wide prediction of topoisomerase IIβ binding by architectural factors and chromatin accessibility

**Pedro Manuel Martínez-García**[1]*, **Miguel García-Torres**[2], **Federico Divina**[2], **José Terrón-Bautista**[1], **Irene Delgado-Sainz**[1], **Francisco Gómez-Vela**[2], **Felipe Cortés-Ledesma**[1,3]*

**1** Centro Andaluz de Biología Molecular y Medicina Regenerativa (CABIMER), CSIC-Universidad de Sevilla-Universidad Pablo de Olavide, Seville, Spain, **2** Division of Computer Science, Universidad Pablo de Olavide, Seville, Spain, **3** Topology and DNA breaks Group, Spanish National Cancer Centre (CNIO), Madrid, Spain

* pmmargar@upo.es (PMMG); fcortes@cnio.es (FCL)

**Data Availability Statement:** All relevant data are within the manuscript and its Supporting information files. TOP2B ChIP-seq datasets

## Abstract

DNA topoisomerase II-β (TOP2B) is fundamental to remove topological problems linked to DNA metabolism and 3D chromatin architecture, but its cut-and-reseal catalytic mechanism can accidentally cause DNA double-strand breaks (DSBs) that can seriously compromise genome integrity. Understanding the factors that determine the genome-wide distribution of TOP2B is therefore not only essential for a complete knowledge of genome dynamics and organization, but also for the implications of TOP2-induced DSBs in the origin of oncogenic translocations and other types of chromosomal rearrangements. Here, we conduct a machine-learning approach for the prediction of TOP2B binding using publicly available sequencing data. We achieve highly accurate predictions, with accessible chromatin and architectural factors being the most informative features. Strikingly, TOP2B is sufficiently explained by only three features: DNase I hypersensitivity, CTCF and cohesin binding, for which genome-wide data are widely available. Based on this, we develop a predictive model for TOP2B genome-wide binding that can be used across cell lines and species, and generate virtual probability tracks that accurately mirror experimental ChIP-seq data. Our results deepen our knowledge on how the accessibility and 3D organization of chromatin determine TOP2B function, and constitute a proof of principle regarding the *in silico* prediction of sequence-independent chromatin-binding factors.

## Author summary

Type II DNA topoisomerases (TOP2) are a double-edged sword. They solve topological problems in the form of supercoiling, knots and tangles that inevitably accompany genome metabolism, but they do so at the cost of transiently cleaving DNA, with the risk that this entails for genome integrity, and the serious consequences for human health, such as neurodegeneration, developmental disorders or predisposition to cancer. A comprehensive analysis of TOP2 distribution throughout the genome is therefore essential for

generated in this study are available under GEO accession number GSE141528 https://www.ncbi.nlm.nih.gov/geo/query/acc.cgi?acc=GSE141528). The code used to train our models and generate genome wide predictions is freely available at https://gitlab.com/mgarciat/genome-wide-prediction-of-topoisomerase-iibeta-binding.

**Funding:** This research was supported by grants from the Spanish Government and the European Regional Development Fund (SAF2017-89619-R, FCL, and TIN2015-64776-C3-2-R, FD), and the European Research Council (ERC-CoG-2014-647359, FCL). CABIMER is supported by the Andalusian Regional Government (Junta de Andalucía). The funders had no role in study design, data collection and analysis, decision to publish, or preparation of the manuscript.

**Competing interests:** The authors have declared that no competing interests exist.

a deep understanding of its function and regulation, and how this can affect genome dynamics and stability. Here, we use machine learning to thoroughly explore genome-wide binding of TOP2B, a vertebrate TOP2 paralog that has been linked to genome organization and cancer-associated translocations. Our analysis shows that TOP2B-DNA binding can be accurately predicted exclusively using information on DNA accessibility and binding of genome-architecture factors. We show that such information is enough to generate virtual maps of TOP2B binding along the genome, which we validate with *de novo* experimental data. Our results highlight the importance of TOP2B for accessibility and 3D organization of chromatin, and show that computationally predicted TOP2 maps can be accurately obtained using minimal publicly available datasets, opening the door for their use in different organisms, cell types and conditions with experimental and/or clinical relevance.

## Introduction

Type II DNA topoisomerases are unique in their ability to catalyze duplex DNA passage, and therefore the only enzymes capable of dealing with superhelical DNA structures, as well as unknotting and decatenating DNA molecules [1]. This places them in a privileged position to integrate and coordinate many aspects of genome organization and metabolism, and have indeed been related to virtually all aspects of chromatin dynamics [2, 3]. While the genomes of invertebrates and lower eukaryotic systems code for only one type II topoisomerase (TOP2), vertebrates encode two close paralogs (TOP2A and TOP2B) with similar catalytic and structural properties but with different roles [2, 4]. TOP2A is essential for cellular viability [5] and is mainly expressed in dividing cells [6], where it is required for chromosome segregation [2]. TOP2B, in contrast, is not essential at a cellular level [7], but is ubiquitously expressed and required for organismal viability due to roles in the transcriptional regulation of genes required for proper development of the nervous system [2, 6, 8].

Consistent with regulatory functions in transcription, TOP2B significantly associates with promoters and DNase I hypersensitivity sites, as well as actively regulated epigenetic modifications and enhancers [9–13]. Furthermore, recent studies have linked TOP2B with 3D genome organization due to a strong colocalization with architectural factors such as CTCF and the cohesin complex protein RAD21 at the boundaries of topologically associated domains (TADs), which has been interpreted as TOP2B functions in resolving topological problems derived from the active organization of the genome in the context of the loop extrusion model [11, 12]. Strikingly, TOP2B and RAD21 are spatially organized around CTCF binding sites, forming ordered triple sites that flank the boundaries of TADs in a way that TOP2B is positioned externally and cohesin internally to the domain loop.

Despite these fundamental roles in the organization and dynamics of the genome, TOP2 function can also pose a threat to its integrity [4]. Aberrant activity of TOP2, which can occur spontaneously or as a consequence of cancer chemotherapy, results in the formation of DNA double-strand breaks (DSBs). These are highly cytotoxic lesions that, if inefficiently or aberrantly repaired, can seriously compromise cell survival and genome stability. Indeed, unrepaired or misrepaired DSBs can lead to genome rearrangements associated with neurodegeneration, sterility, developmental disorders and predisposition to cancer [14–16]. In particular, transcription-associated TOP2B function at TAD boundaries has been recently linked to oncogenic chromosomal translocations [17, 18]. The specific functions of TOP2B in these processes are, however, still largely unexplored and far from being fully understood. A deep

understanding of TOP2B function and regulation in a genome-wide context, and how it can result in chromosome fragility and alterations, will require thorough and unbiased studies, integrating different systems, cellular types and conditions.

In this work, we take advantage of the wealth of high-throughput sequencing data to comprehensively study the informative power of a wide set of chromatin features to predict TOP2B binding. We show that TOP2B localization can be accurately predicted using publicly available sequencing data, and identify architectural and open chromatin factors as the most informative features, in agreement with previously reported data [11, 12]. Moreover, feature selection analysis indicates that TOP2B can be faithfully predicted using only three sequencing datasets. Indeed, models trained with DNase-seq and ChIP-seq of CTCF and RAD21 allow for accurate prediction across different mouse systems, supporting a generalizing association of TOP2B with these features. Finally, to validate our model we generate predictions in mouse thymocytes and human MCF7 cells and compare them with ChIP-seq data obtained in our laboratory. Genome wide predictive tracks accurately mirror experimental TOP2B tracks in both organisms, showing that our model can be used to generate virtual TOP2B signals in potentially any mammalian cell type and condition for which sequencing data of CTCF, RAD21 and DNase I hypersensitivity are available.

## Materials and methods

### Processing publicly available data

The experimental data used in this study is summarized in S1 Table. When available, we batch-downloaded mm9 and hg19 BAM files from ENCODE [19]. Raw sequencing data was processed by first quality filtering and merging the reads of biological replicates and corresponding input samples (in the case of ChIP-seq experiments). Then we mapped them to the mouse or human genome (mm9 or hg19) using Bowtie 1.2 [20] with option "-m 1" so that reads that mapped only once to the genome were retained. Genome track plots of Fig 1 and S1 Fig were generated using Bioconductor packages Gviz [21] and trackViewer [22].

### Peak calling and generation of randomized regions

To generate our predictive models, we identified TOP2B binding sites in the mouse learning systems (liver, MEFs and activated B cells) using MACS2 [23] with option "-q 0.01" and keeping only peaks with a fold change over the control sample greater than 5. Then, peaks overlapping >20% of their width with non-mappable regions or regions of the ENCODE blacklist [24] were discarded. Since we were interested in highly enriched TOP2B binding sites to ensure robust learning, peaks were called for individual replicates (when available) and only overlapping peaks were kept. For each training system, we generated the same number of random regions in the genome than TOP2B binding sites were identified. Such regions were searched to have 300 bp length and were selected so that they had the same distribution across chromosomes as TOP2B peaks. Again, non-mappable regions and regions of the ENCODE blacklist were discarded. To make sure they did not overlap TOP2B binding sites not detected by MACS, regions with high levels of TOP2B/INPUT ChIP-seq signal were also discarded. In order to account for sequence composition biases, an additional set of random regions was generated so that their distribution of sequence G+C content matched that of TOP2B peaks. We proceeded using an iterative approach. Briefly, the G+C content of TOP2B peaks was first computed and an initial set of random regions was generated for each chromosome. Random regions having similar G+C content than any TOP2B peak were retained while matched TOP2B peaks were not considered in the next iteration. This process was repeated until the number of GC-random regions and TOP2B peaks were equal for each chromosome.

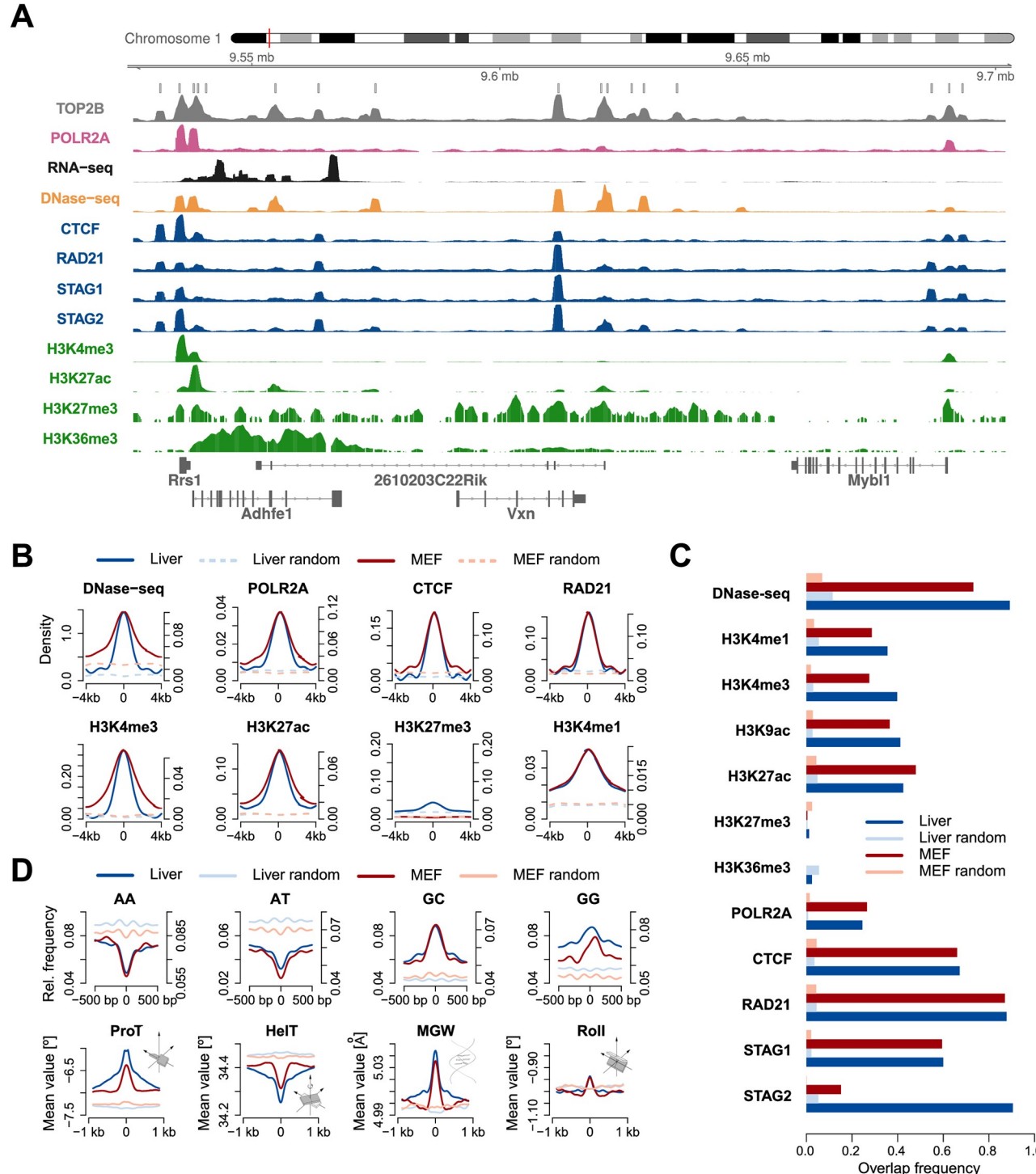

**Fig 1. Chromatin landscape of TOP2B. A**. Genome browser view for TOP2B ChIP-seq signal and other relevant chromatin features in mouse liver. **B**. Average reads enrichment of selected chromatin features within ±4 kb of TOP2B binding center (solid lines) and random regions (dashed lines) in mouse liver (blue) and MEFs (red). Signals were smoothed using nucleR [50]. Left and right sided ordinate axes correspond to mouse liver and MEFs, respectively. **C**. Overlap frequencies of chromatin features at TOP2B binding sites as compared to random regions in mouse liver and MEFs. **D**. Top panel shows relative frequency of several dinucleotides within ±500 bp of TOP2B binding center and random regions. Bottom panel shows average of 3D DNA shape features within ±1 kb of TOP2B binding center and random regions. Profiles corresponding to MEFs and liver are colored as in C.

When we processed the experimental ChIP-seq samples generated in our lab, we found that MACS2 greatly underestimated the number of TOP2B binding sites. For this reason, we used HOMER [25] to identify peaks in our samples. Libraries were first subsampled so that both the test and the control samples had the same number of reads. Then, HOMER was called with option "-style factor". Only peaks with a fold change over the control sample (IgG) greater than 5 and not overlapping non-mappable regions and regions of the ENCODE blacklist were retained.

## Distribution of TOP2B binding sites across the genome

TOP2B binding sites and random genomic regions were annotated using two different approaches. On the one hand, peaks were classified relative to the mouse TSSs (transcription start sites) according to the mm9 transcripts annotation of UCSC [26] using ChIPseeker [27]. In addition, we grouped peaks into five classes as described in Matthews and Waxman (2018). Briefly, open chromatin regions (DNase-seq peaks) were identified and classified based on H3K4me1, H3K4me3, and CTCF ChIP-seq signals. Promoters were defined as peaks with > 1.5 H3K4me3/H3K4me1 fold ratio. Enhancers were defined as peaks with < 0.67 H3K4me3/H3K4me1 fold ratio. The remaining DNase peaks were separated into those with a roughly equal H3K4me3/H3K4me1 fold ratio and those with low signal for both histone marks. From the the former set, those overlapping TSSs were classified as weak promoters. Low signal peaks that overlapped CTCF binding sites were classified as insulators. Finally, the remaining low signal peaks not overlapping TSSs were classified as weak enhancers. For a more detailed description, see [28].

## Quantification of high-throughput sequencing data

To score sequencing data around TOP2B binding sites, we based on the scoring approach of [29]. For ChIP-seq experiments, reads aligned to a 300 bp window centered on TOP2B binding center were first considered, normalized to the size of the corresponding sequencing library and scaled to the size of the smaller library (among test and input). Then, they were added 1 as a pseudocount (to avoid undefined values in posterior logarithmic transformations) and the ratio of test versus input signal was calculated. Finally, this value was log transformed. RNA-seq, DNase-seq and MNase-seq were scored as the number of reads aligned within the same 300 bp window divided by the library size. CpG methylation was measured as the total percentage of methylated CG for each CpG dinucleotide as described in [30].

## DNA shape and sequence features

Three-dimensional DNA shape features at base pair or base pair step resolution were derived from the high-throughput approach implemented in the Bioconductor library DNAshapeR [31]. We used thirteen DNA shape features: helix twist, propeller twist, minor groove width, roll, shift, slide, rise, tilt, shear, stretch, stagger, buckle and opening [32]. For a given DNA sequence of length L, the k-mer feature at each nucleotide position was encoded as a binary vector of length $4^k$, as described in [33]. A value of 1 represents the occurrence of a particular k-mer starting at that position. The k-mer features for a whole DNA sequence were then encoded by the concatenation of the k-mer feature vectors at each position of the sequence. As a result, we obtained a binary vector of length $4^k(L-k+1)$.

## Machine learning

**Training data matrix.** After scoring the sequencing experiments as well as DNA shape and sequence parameters, we built up a final data matrix with rows representing TOP2B/

random sites and columns corresponding to scored features. Since we identified a total of 13, 128 and 8, 413 TOP2B peaks in mouse liver and MEFs respectively, we ended up with model matrices of 32, 766 columns and either 26, 256 (mouse liver) or 16, 826 (MEFs) rows.

**Classifiers.** The Naive Bayes [34] (NB) classifier applies the Bayes rule to compute the probability of a class label given an instance. For such purpose, the classification model learns, from training data, the conditional probability of each feature given the class label. NB assumes that all features are conditionally independent given the values of the class.

Support Vector Machine [35] (SVM) is a classifier based on the idea of finding a hyperplane that optimally separates the data into two categories by solving an optimization problem. Such hyperplane, is the one that maximizes the margin between classes. In case of non-separable data, it optimizes a weighted combination of the misclassification rate and the distance of the decision boundary to any sample vector. After several tests with different kernels (linear, polynomial, radial basis and sigmoid), we use a linear kernel due to its good performance and simplicity.

Random Forests [36] is a widely applied tree-based supervised machine-learning algorithm designed as a combination of bagging and a random selection of features that has proven to be an effective tool for the statistical analysis of high-dimensional genomic data [37].

**Feature selection algorithms.** Fast Correlation Based Filter [38] (FCBF) is an efficient heuristic that, based on information theory measures, performs a relevance and redundancy analysis for selecting a good subset of features. Basically, FCBF is a backward search strategy that uses Symmetrical Uncertainty (SU) as goodness function to measure non-linear dependencies between features. In order to identify relevant features, it requires to set a threshold parameter $\delta$ so that only values above $\delta$ are considered relevant. In this work we set $\delta = 0$ since there is no rule about this parameter tuning and in the datasets under study only a small subset of features have a SU value different to 0.

Scatter Search [39, 40] (SS) is a population-based metaheuristic that uses intensification and diversification mechanisms to generate new solutions. The strategy starts by generating an initial population of diverse solutions from the solution space. Then, it selects a subset of features, called Reference Set (RefSet), of high quality and dispersed solutions from the initial population. In the next step the method selects all subsets consisting of two solutions and combine them in order to generate new solutions that are improved by applying a local search which will yield to local optima. Finally, a static update of the reference set is carried out selecting solutions according to quality and diversity from the union of the original RefSet and the new local optima found. As goodness of feature subsets, the Correlation Feature Selection [40] (CFS) is used.

In this work we use the Correlation Feature Selection [41] (CFS) measure as goodness function. CFS evaluates subsets of features by considering the non-linear correlation Symmetrical Uncertainty function.

## Genome wide predictions

Once we identified the most predictive features, a generalized TOP2B binding model was built by training on DNase-seq, RAD21 and CTCF data from mouse liver, MEFs and activated B cells using Random Forests [36]. We first used 5-fold cross-validation to confirm that the model was able to accurately predict TOP2B in the training systems (as shown in Results and Discussion) and then applied the model to the mappable genomes of both the training and the test systems (mouse thymus and human MCF7). First, we split the genomes into bins of 300 bp with sliding windows of 50 bp. Then we scanned each bin with our model and obtained a

TOP2B binding probability vector that was used to build up a bedgraph file. Probability track plots were generated using the UCSC genome browser [26].

## ChIP-seq

Cells were fixed by adding formaldehyde to a final concentration of 1% and incubated at 37˚C for 10 min. Fixation was quenched by adding glycine to a final concentration of 125 mM at cell culture plates. After two washes with cold PBS, in the presence of complete protease inhibitor cocktail (Roche) and PMSF, cell pellet was lysed in two steps using 0.5% NP-40 buffer for nucleus isolation and SDS 1% lysis buffer for nuclear lysis. Sonication was performed using Bioruptor (Diagenode, UCD-200) at high intensity and three cycles of 10 min (30" sonication, 30" pause) and chromatin was clarified by centrifugation (17000xg, 10 min, 4 ˚C). For IP, 30 μg chromatin and 4 μg antibody (anti-TOP2B, SIGMA-HPA024120) were incubated o/n in IP buffer (0.1% SDS, 1% TX-100, 2 mM EDTA, 20 mM TrisHCl pH8, 150 mM NaCl) at 4˚C, and then with 25 μL of pre-blocked (1 mg/ml BSA) Dynabeads protein A and Dynabeads protein G (ThermoFisher) for 4 hours. Beads were then sequentially washed with IP buffer, IP buffer containing 500 mM NaCl and LiCl buffer (0.25 M LiCL, 1% NP40, 1% NaDoc, 20 mM TrisHCl pH8 and 1 mM EDTA). ChIPmentation was carried out as previously described (Schmidl 2015) using Tagment DNA Enzyme provided by the Proteomic Service of CABD (Centro Andaluz de Biología del Desarrollo). DNA was eluted by incubation at 50˚C in 100 $\mu L$ elution buffer (1% SDS, 100 mM NaHCO3) for 30 min followed by an incubation with 200 mM NaCl and 10 μg of Proteinase K (ThermoFisher) to revert cross-linking. DNA was then purified using Qiagen PCR Purification columns. Libraries were amplified for N-1 cycles (being N the optimum Cq determined by qPCR reaction) using NEBNext High-Fidelity Polymerase (M0541, New England Biolabs), purified and size-selected using Sera-Mag Select (GE Healthcare) and sequenced using Illumina NextSeq 500 in a single-end configuration. ChIP-seq datasets corresponding to two replicates and one IgG control sample are available under GEO accession number GSE141528.

## Western blot analysis

For protein extraction, cell pellets were resuspended in RIPA buffer supplemented with protease inhibitors and then sonicated, clarified and loaded after quantification in a 4-20% Mini-PROTEAN tris-Glycine Precast Protein Gels (Biorad, 4561096). Gels were electroblotted onto Immobilon-FL Transfer Membrane (Millipore). Western blot was performed following standard protocol for Odyssey CLx analysis (LI-COB BIOSCIENCES, Lincoln, NE) and for both anti-TOP2B antibodies (SIGMA, HPA024120 and NOVUS, NB100-40842) 1:3000 dilution of the antibody was used in Odyssey Blocking Buffer.

## Code availability

The code used to train our models and generate genome wide predictions is freely available at GitLab.

# Results and discussion

## Chromatin landscape of TOP2B binding sites

We started by comprehensively assessing the chromatin landscape of TOP2B binding sites in two mouse systems: embryonic fibroblasts (MEFs) and liver. We collected high-throughput sequencing data from ENCODE [19] and several independent studies [11, 12, 42–49] (S1 Table) and observed the expected colocalization with open chromatin sites, RNA polymerase

II (Pol2), architectural components and transcription associated histone modifications; genome tracks in both systems are provided as an example (Fig 1A and S1 Fig). Then, we identified enriched TOP2B peaks and studied the position specific distribution of each chromatin feature around the TOP2B binding center (Fig 1B). Several marks were found to be highly enriched in both systems, such as DNase-seq, CTCF and the cohesin complex members STAG1, STAG2 and RAD21. These features also showed a high colocalization frequency with TOP2B (Fig 1C), in line with previous findings [11, 12]. For instance, 89% of TOP2B sites in mouse liver colocalized with DNase-seq peaks, while for randomly selected regions this percentage decreased to 11%. Similarly, 87% of TOP2B peaks overlapped with RAD21 in MEFs as compared to 4% of random regions. These observations agree with a preferential association of TOP2B with accessible chromatin, actively transcribed promoters and regulators of genome architecture [10–12], and suggest that chromatin features measured by next generation sequencing could provide valuable information for the prediction of TOP2B binding.

Current experimental data suggest that TOP2B does not bind to a specific DNA motif [11], which agrees with topoisomerases specifically acting where DNA topological problems arise. However, TOP2B associates with DNA regions that are frequently bound by sequence-specific transcription factors [9–11], so the fact that particular topological structures can be favored by specific properties of DNA sequence is still a possibility. Indeed, a recent study has provided evidence that DNA sequence governs the location of supercoiled DNA [51]. To explore whether sequence composition of TOP2B binding sites can be used in our predictive approach, we analyzed the spatial distribution of DNA sequence dinucleotides within 1kb windows centered on TOP2B peaks. For both murine systems, TOP2B sites showed an increase of GC and GG dinucleotides while AT and AA were depleted (Fig 1D, top). This result agrees with previously reported GC-rich motifs, such as CTCF and ESR1, at TOP2B occupied regions [10, 11].

Another feature that could potentially help in the prediction of TOP2B binding is DNA shape, which has been shown to affect the binding preferences of a number of DNA associated proteins [33, 52, 53]. In this sense, models trained with information on DNA shape have demonstrated significant improvements in transcription factor binding predictions [33]. To further inspect whether the observed sequence preferences of TOP2B sites are accompanied by specific 3D conformational parameters, we derived high-throughput predictions of DNA shape features from TOP2B peaks. Position specific profiles of such features revealed a specific pattern around the TOP2B binding center, characterized by decreased helix twist, increased minor groove width and propeller twist, and a modest enrichment of roll (Fig 1D, bottom). Such conformation suggests helical unwinding as consequence of a decrease in helix twist, which together with the widening of the minor groove could be providing an energetically favorable scenario for TOP2B binding.

Finally, we also interrogated whether CpG methylation is informative for TOP2B binding prediction. CpG methylation is a well-known epigenetic mark that has proven predictive ability for the localization of DNA binding proteins [54–56]. We profiled whole genome bisulfite sequencing (WGBS) data around TOP2B binding sites and found decreased CpG methylation as compared to random regions (S2 Fig), showing that this feature is likely to be informative for TOP2B prediction.

## A predictive model for TOP2B binding based on chromatin features

In order to assess whether chromatin features could discriminate TOP2B binding sites from the rest of the genome, we applied the computational approach outlined in Fig 2. For both MEFs and liver, we considered peaks identified in the previous section, resized them to 300 bp and generated the same number of random genomic regions (see Materials and methods).

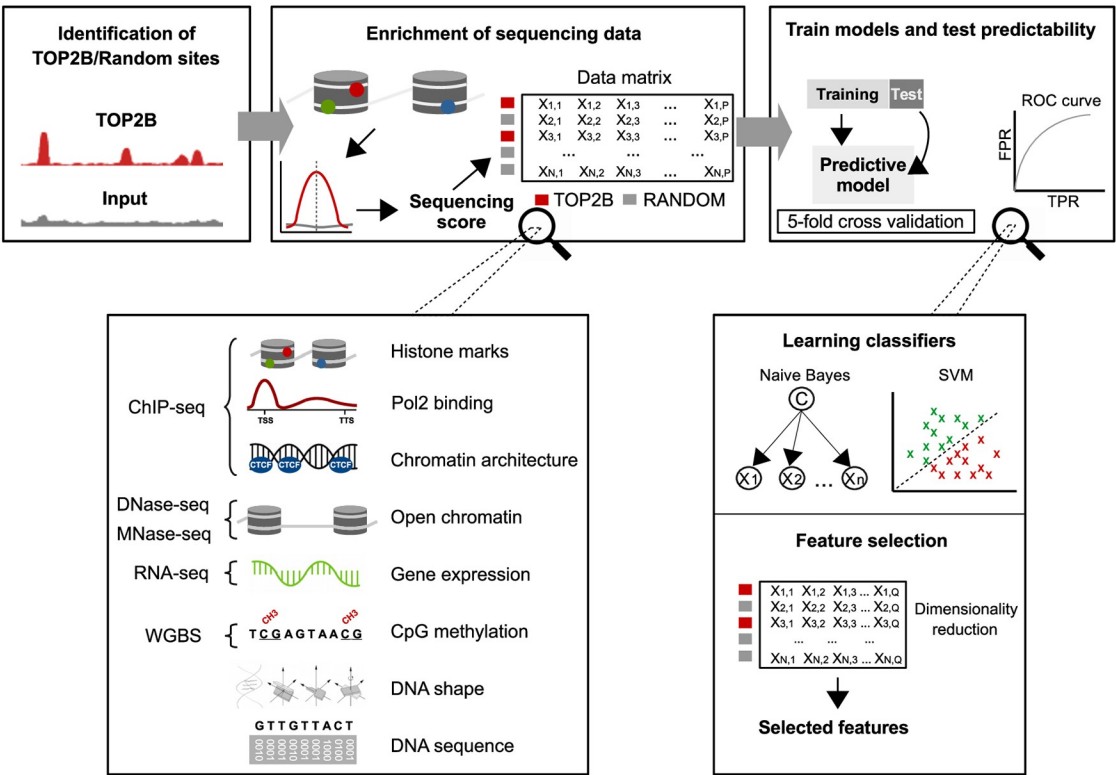

**Fig 2. Machine learning schema for the prediction of TOP2B binding.** TOP2B binding sites and random regions were first identified. Then, 15 high-throughput sequencing experiments together with DNA sequence and shape features were scored around such regions, which resulted a data matrix with rows representing TOP2B/random sites and columns representing the scored features. Finally, binary classifiers were trained and tested using 5 fold cross-validation and feature selection was applied to identify the most informative features.

Then we scored 15 high-throughput sequencing experiments (S1 Table) together with DNA sequence and shape features within such regions. Regarding DNA sequence, we represented DNA 1-mers, 2-mers and 3-mers for each nucleotide position in the TOP2B binding sites as described in [33] (see Materials and methods). This produced 1, 200 parameters representing 1-mers, 4, 784 parameters representing 2-mers and 19, 072 parameters representing 3-mers. We also included information on 13 DNA shape features using DNAshape method [57], which added other 7, 695 parameters to the model. This parametrization allowed us to measure the predictive ability of DNA sequence and shape to explain TOP2B binding with an unprecedented resolution. As a result, we ended up with model matrices of 32, 766 columns and either 13, 128 (liver) or 8, 413 (MEFs) rows. Finally, binary classifiers were trained and tested using 5-fold cross-validation.

Since TOP2B binding regions display increased G+C content as compared to the average of the mouse genome, we trained one more model in which random regions were selected so that their distribution of sequence G+C content matches that of TOP2B peaks. This allowed us to account for potential biases of the predictions due to differences in G+C content. The final sets of TOP2B, random and GC-corrected random peaks displayed the expected genomic distribution [9–13], with TOP2B binding sites showing higher co-localization with promoters, enhancers and insulators than randomized regions (S3(A) and S3(B) Fig). Nevertheless, forcing the G+C content of random regions to match that of TOP2B binding sites led to the former being located at GC rich sites, which in turn made them show a moderate co-localization with

**Table 1. Performance of SVM and NB classifiers in MEF.**

| System | Category | GC corrected | | Random | |
|---|---|---|---|---|---|
| | | NB | SVM | NB | SVM |
| MEF | All | 74.18 ± 1.08 | 96.72 ± 0.18 | 70.94 ± 0.75 | 98.06 ± 0.28 |
| | Sequence | 59.15 ± 0.61 | 55.44 ± 1.10 | 66.87 ± 0.88 | 64.41 ± 0.48 |
| | 3D shape | 55.08 ± 0.92 | 54.32 ± 0.87 | 66.23 ± 0.54 | 61.80 ± 0.97 |
| | Histone marks | 84.82 ± 0.73 | 83.90 ± 0.66 | 88.61 ± 0.70 | 87.57 ± 0.74 |
| | Architecture | 90.35 ± 0.61 | 94.98 ± 0.32 | 93.15 ± 0.58 | 96.52 ± 0.39 |
| | Open chromatin | 81.92 ± 0.58 | 83.96 ± 1.44 | 90.48 ± 0.61 | 92.35 ± 0.63 |
| | Expression | 50.36 ± 0.25 | 50.48 ± 0.54 | 50.77 ± 0.22 | 52.52 ± 1.57 |
| | Pol2 | 60.60 ± 0.62 | 61.48 ± 1.06 | 57.71 ± 0.46 | 56.09 ± 0.92 |
| | CpG methylation | 69.45 ± 0.88 | 73.82 ± 0.88 | 63.98 ± 0.79 | 68.90 ± 1.09 |

actively regulated regions. Since TOP2B is expected to localize at such regions, we generated heatmaps to examine the ChIP-seq signal at TOP2B, random, and GC-corrected random peaks and observed enrichment only at the center of TOP2B peaks (S3(C) Fig), indicating that our background regions are adequate for subsequent training.

We employed Support Vector Machine (SVM) [35] and Naive Bayes (NB) [34] to build up the predictive models and used 5-fold cross-validation to estimate the prediction accuracy. We started by training on the whole set of features. Accurate predictions were obtained for the two murine systems using both the GC-corrected and the regular model (Tables 1 and 2; Fig 3 and S4 Fig), indicating that chromatin features can faithfully explain TOP2B binding. As a matter of fact, SVM trained on all features performed considerably better than NB. For instance, while regular models trained with NB achieved accuracies of 71% and 77.4% for MEF and liver, respectively, these values increased to 98% and 97.4% when using SVM. Similarly, accuracies obtained training the GC-corrected models with NB were 74.1% and 76.2% for MEF and liver, respectively, whereas SVM produced accuracies of 96.7% and 93%. Such performance differences are likely to be explained by many features showing dependencies to each other or being poorly informative. This would impair NB performance, which assumes all features are independent. Indeed, removal of not informative features leads to a significant increase of accuracies achieved by the NB algorithm (see next sections).

**Table 2. Performance of SVM and NB classifiers in mouse liver.**

| System | Category | GC corrected | | Random | |
|---|---|---|---|---|---|
| | | NB | SVM | NB | SVM |
| Liver | All | 76.20 ± 0.97 | 93.00 ± 0.56 | 77.43 ± 0.20 | 97.47 ± 0.13 |
| | Sequence | 62.95 ± 0.70 | 56.64 ± 1.10 | 74.98 ± 0.31 | 72.78 ± 0.94 |
| | 3D shape | 57.97 ± 0.66 | 54.00 ± 1.42 | 73.19 ± 0.38 | 69.71 ± 0.46 |
| | Histone marks | 78.03 ± 0.32 | 81.17 ± 0.47 | 78.97 ± 0.44 | 85.34 ± 0.41 |
| | Architecture | 87.45 ± 0.53 | 91.74 ± 0.37 | 89.50 ± 0.54 | 94.26 ± 0.56 |
| | Open chromatin | 71.68 ± 0.46 | 85.32 ± 4.22 | 83.35 ± 0.92 | 92.01 ± 1.57 |
| | Expression | 50.00 ± 0.01 | 50.00 ± 0.01 | 50.04 ± 0.28 | 50.02 ± 0.01 |
| | Pol2 | 54.92 ± 0.24 | 53.97 ± 0.42 | 52.95 ± 0.73 | 52.63 ± 0.90 |
| | CpG methylation | 71.58 ± 0.34 | 75.31 ± 0.34 | 66.56 ± 0.36 | 69.58 ± 0.44 |

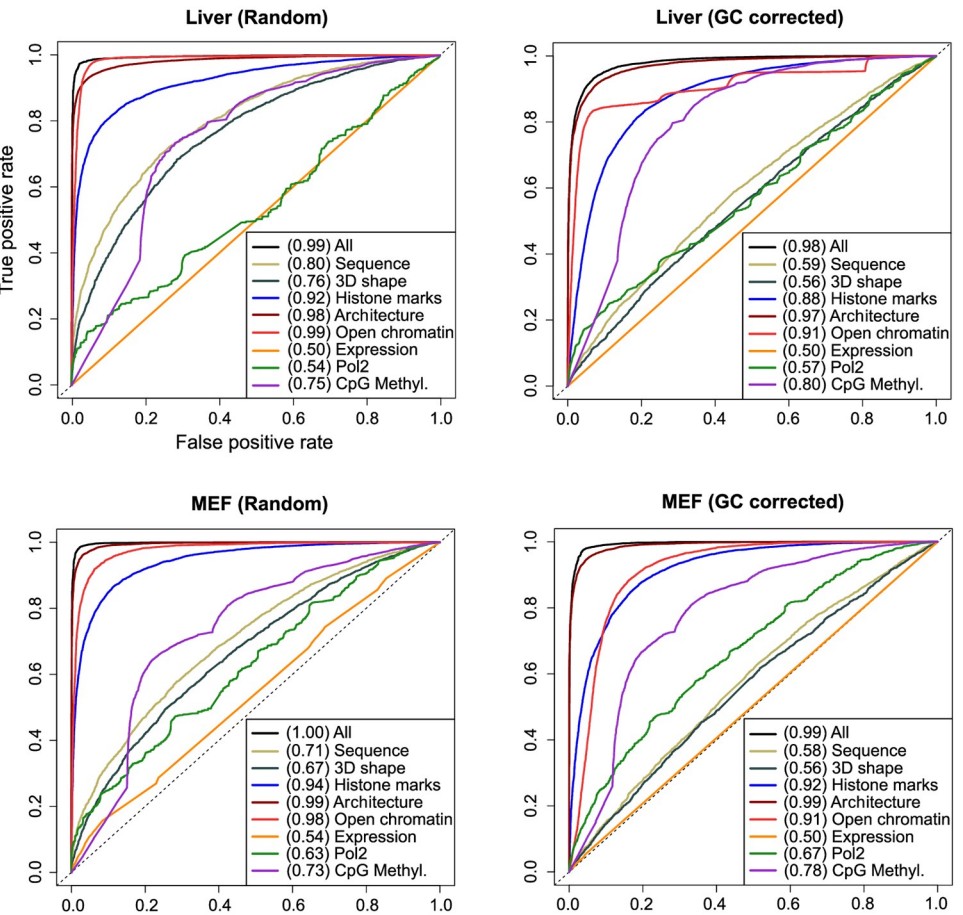

**Fig 3. Chromatin features predict TOP2B binding.** ROC curves and AUC values for Support Vector Machine models trained on the indicated sets of features (for Naive Bayes models, see S4 Fig).

## Chromatin architecture and open chromatin are the most informative features

To explore whether specific molecular events provide different prediction abilities for TOP2B binding, chromatin features were grouped into 8 categories (Fig 2): open chromatin (DNAse-seq, MNase-seq), histone marks (H3K4me1, H3K4me3, H3K9ac, H3K27ac, H3K27me3, H3K36me3), Pol2 binding (POLR2A ChIP-seq), architectural components (ChIP-seq of CTCF and the cohesin complex components RAD21, STAG1 and STAG2), gene expression (RNA-seq), CpG methylation (WGBS), DNA sequence and DNA shape (measured using DNAshapeR [31]). Then, SVM and NB models were constructed for each of such categories.

Different predictive performances were observed for the grouped molecular events, with similar results across systems (Tables 1 and 2; Fig 3 and S4 Fig). Models trained with architectural components achieved the highest prediction accuracies, ranging from 87.4% to 96.5%, with area under ROC curves (AUC) of 0.96—0.99 (Fig 3 and S4 Fig). This is in agreement with previous findings of TOP2B associating with the boundaries of TADs [11, 12]. In the same line, models built using open chromatin factors reflect TOP2B preference to bind accessible DNA, showing accuracies of 71.6%—92.35% and AUC of 0.88—0.99 (Tables 1 and 2; Fig 3 and S4 Fig).

As expected, histone-mark models also obtained reasonably high prediction performances, with accuracies ranging from 78% to 88.6% and AUC of 0.84—0.95 (Tables 1 and 2; Fig 3 and S4 Fig). The set of histone modifications used to feed these models include a combination of promoter, enhancer and gene body associated marks, which represents common genomic regions where TOP2B tends to localize.

In contrast to the above results, gene expression (measured by RNA-seq) appeared to be the least informative feature for TOP2B binding, with accuracies of around 50% and AUC of 0.5—0.54. Interestingly, and despite the observed co-occurrence of TOP2B and Pol2 at gene promoters (Fig 1 and S1 Fig), the model trained with Pol2 binding information also exhibited poor prediction accuracies, ranging from 52.6% to 61.4% and AUC of 0.53—0.67. Although these observations seem to contrast with previous findings of TOP2B being involved in transcriptional regulation [58], we argue that other chromatin features, such as histone marks or DNase-seq may be capturing such association. Given the observed positive correlation of TOP2B-induced DSBs at CTCF and cohesin-bound regions with transcriptional output [17, 18], even architectural factors can be capturing TOP2B association with transcription within our model.

## The impact of correcting for GC-bias

According to the performances achieved, the effect of training models using GC-corrected background regions is negligible for the categories analyzed. As expected, this is not the case for DNA sequence and DNA shape models, which obtained decreased prediction accuracies when trained with GC-corrected sites as compared to those achieved using pure random regions. While sequence content exhibited a prediction accuracy of 55.4% and 62.9% and AUC of 0.58—0.66 for GC-corrected models, these values increased to 64.1%—74.9% and AUC of 0.71—0.84 for random models (Tables 1 and 2; Fig 3 and S4 Fig). Similarly, sequence-derived 3D shape models built with GC-corrected regions achieved accuracies from 54% to 57.9% and AUC of 0.56—0.6, whereas pure random models obtained accuracies of 61.8%—73.1% and AUC of 0.67—0.78. Given the remarkable resolution of our model in measuring DNA sequence and sequence-derived 3D DNA shape, these results suggest that sequence information alone is only poorly informative of TOP2B binding, and strengthen the idea of topoisomerases acting where DNA topological problems arise.

The predictive performances of models trained with CpG methylation data were also influenced (although only moderately) by the use of GC-corrected sites as background regions. While pure random models achieved modest accuracies, ranging from 64% to 69.5% and AUC of 0.73—0.76, GC-corrected models obtained moderately higher accuracies of 69.4%—75.3% and increased AUC of 0.78—0.81 (Tables 1 and 2; Fig 3 and S4 Fig). When forcing the G+C content of random regions to match that of TOP2B binding sites, they tend to be located at GC rich sites not occupied by TOP2B. Such regions happen to be hypermethylated as compared with TOP2B sites (S2 Fig), which leads to an increased predictive power of CpG methylation.

Given the modest differences found using random or GC-corrected random regions in model performances, from this point on only GC-corrected random regions were considered.

## Three chromatin features explain TOP2B binding genome wide

With the aim of identifying the most relevant features to explain TOP2B binding, we applied feature selection, which reduces the number of attributes in the data matrix while trying to keep (or even improve) the predictive power of the classifier. We used two feature selection algorithms using 5-fold cross-validation: Fast Correlation Based Filter (FCBF) [38] and Scatter

**Table 3. Performance of SVM and NB classifiers using the features selected by FCBF and SS strategies.**

| System | Algorithm | NB | SVM | #features | Stability |
|--------|-----------|-----|------|-----------|-----------|
| MEF | FCBF | 94.60 ± 0.35 | 96.11 ± 0.28 | 46.80 ± 3.90 | 0.113 |
| | SS | 93.88 ± 0.31 | 95.51 ± 0.33 | 3.00 ± 0.00 | 1 |
| Liver | FCBF | 92.62 ± 0.51 | 92.84 ± 1.91 | 22.20 ± 4.38 | 0.132 |
| | SS | 92.90 ± 0.44 | 92.10 ± 1.97 | 3.00 ± 0.00 | 1 |

Search (SS) [39, 40] (see Materials and methods). In addition to the accuracy and the AUC values, here we also considered the average number of features selected by each strategy and the stability of each algorithm. The stability (or robustness) evaluates the sensitivity to variations of a feature selection algorithm in the dataset. This issue is especially important in high dimensional domains where strategies may yield to different solutions with similar performance and, therefore, this measure enhances the confidence in the analysis of the results.

Both feature selection algorithms succeeded in finding a subset of features that faithfully explained TOP2B binding in both systems, with accuracies ranging from 92.1% to 96.11% (Table 3). As a matter of fact, although in mouse liver the two classification approaches (SVM and NB) achieved the same performance for both feature selection algorithms, this was not the case in MEFs, in which the SVM slightly outperformed NB predictions. Regarding the number of selected features, while FCBF needed averages of 46 and 22 features to optimize the prediction of TOP2B in MEF and liver, respectively, SS only needed 3 features in both systems (S2 Table). It is worth stressing that SS always finds, in every iteration, the same 3 features, achieving a stability score of 1 (Table 3). On the other hand, features selected by FCBF vary from different iterations, which confers a poor robustness score to this strategy.

The most selected features for each strategy are shown in Fig 4A. For each histogram and feature, the white bar height indicates the frequency of selection and the black bar height is the Symmetrical Uncertainty (SU) value with respect to the class (TOP2B). SU [59] is a widely used technique that allows to measure the relevance between two variables and can be interpreted as a correlation measure to identify non-linear dependencies between features. Due to the mentioned low stability score obtained with the FCBF approach, only 5 out of the average 46 and 3 out of the average 22 features were selected in every iteration for MEFs and liver, respectively (S2 Table). In MEFs, such 5 features were RAD21, DNase-seq, WGBS, MNase and the DNA shape associated translational parameter Shift. In liver, the three selected features corresponded to DNase-seq, STAG2 and WGBS. It is worth mentioning that the only features selected in 4 of the 5 iterations were the DNA shape associated parameters Slide and Shear in MEFs and Helix Twist in liver. The recurrent selection of such DNA shape features in both systems may indicate a modest contribution of these parameters to TOP2B binding. In any case, most of the predictive power was sustained by open chromatin features (DNAse-seq, MNase-seq), cohesin complex associated factors (RAD21, STAG2) and to a lesser extent CpG methylation (WGBS).

A significantly more robust and consistent selection was achieved when using SS (Fig 4A), where only three features were selected in every iteration for both systems. Furthermore, DNase-seq and RAD21 were both selected for MEFs and liver, which is in line with above results and highlights the influence of accessible chromatin and cohesin factors for TOP2B binding across systems. As a matter of fact, while in MEFs SS selected CTCF at every iteration, this was not the case in liver, where the cohesin complex factor STAG2 was always selected (Fig 4A).

A summary of the features selected by FCBS and SS in both systems is shown in Fig 4B. Top and bottom aligned dots indicate selection of a given feature by the corresponding selection

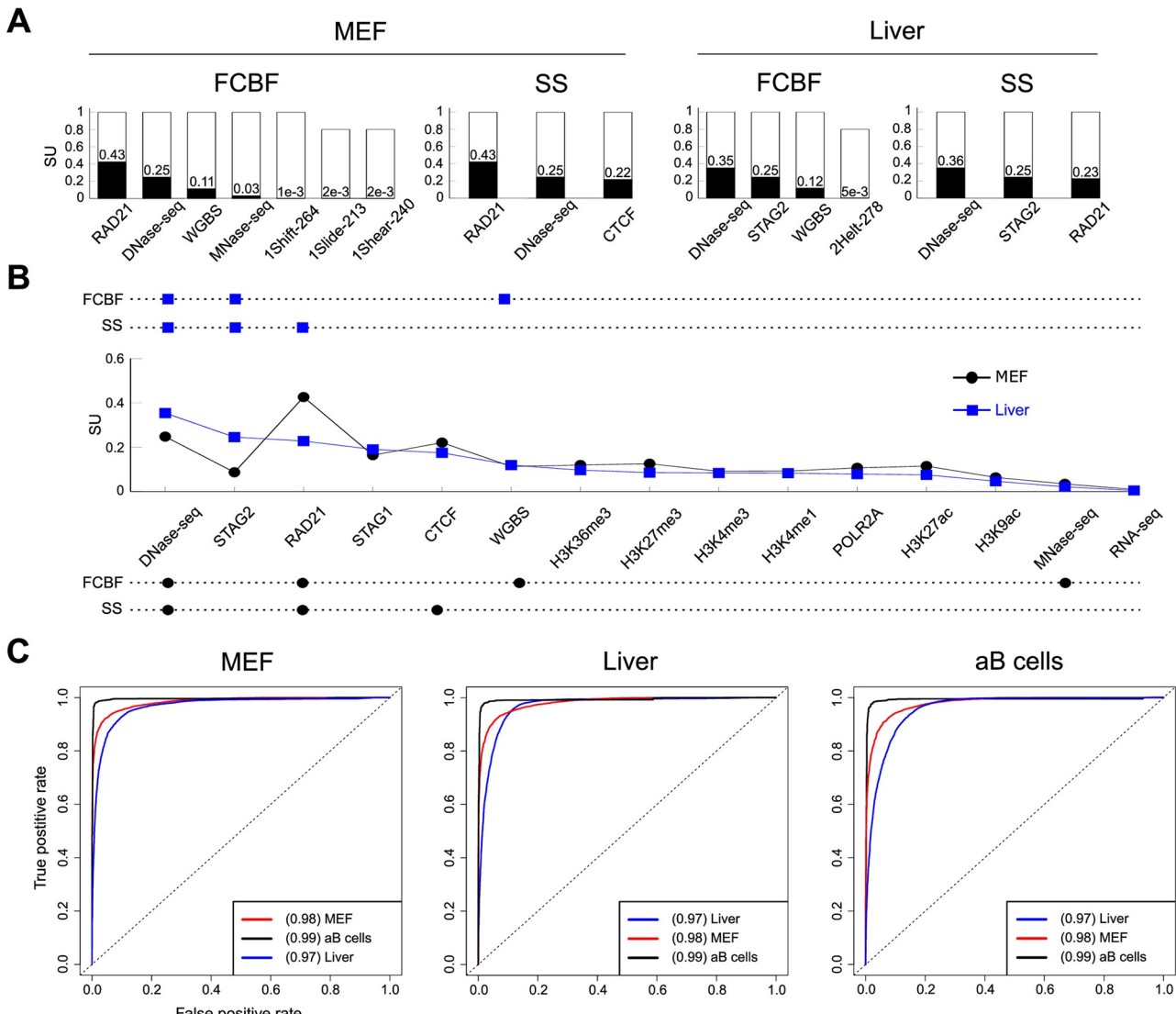

**Fig 4. Three features accurately predict TOP2B. A**. Top features selected by Fast Correlation Based Filter and Scatter Search algorithms. For each histogram and feature, the white bar height indicates the frequency of selection and the black bar height is the Symmetrical Uncertainty (SU) value with respect to the class (TOP2B). Indexes of DNA shape parameters indicate position within the corresponding parameter vector associated to the 300 bp width of modeled TOP2B binding sites (see Materials and methods). **B**. Summary of the most selected features by both algorithms. Top and bottom aligned dots indicate selection of a given feature by the corresponding selection algorithm in liver and MEFs, respectively. In the middle, the SU of each feature is displayed. Only the top fifteen features according to their SU are shown, which happen to match in both systems. **C**. ROC curves and AUC values for Naive Bayes models trained on either MEF, liver or activated B cells and applied to the three systems (for Support Vector Machine and Random Forests models, see S4 Fig). Only DNase-seq, RAD21 and CTCF binding data were used for training.

algorithm in liver and MEFs, respectively. In the middle, the relevance (SU) of each feature with respect to TOP2B is displayed. Only the top fifteen features according to their SU are shown, which happen to match in both systems. With the exception of STAG2, the trends of liver and MEFs are very similar, indicating that the specific contributions of genomic features for TOP2B prediction are conserved across systems. Furthermore, the results obtained by the SS algorithm strengthen the importance of accessible chromatin and architectural factors for TOP2B binding, and shows that a small number of these features are enough to achieve accurate genome-wide predictions. Indeed, comparison of models trained using the whole set of

chromatin features with models trained only with SS-selected features showed that both sets achieved similar performances (S5 Fig). Therefore, we decided to use only the three SS-selected features for subsequent analyses. It is noteworthy to mention that, since data on STAG2 binding is rather scarce, we focused on DNase-seq, RAD21 and CTCF. Although the latter was only selected in MEFs, it was among the most informative features in mouse liver (Fig 4B; S2 Table), and there is a wealth of ChIP-seq data available for this protein. Furthermore, comparison of DNase-RAD21-CTCF and DNase-RAD21-STAG2 models showed that both approaches achieve similar performances in this tissue (S6 Fig and S3 Table), and consequently, we did not loose predictive power.

## Prediction across systems

We next asked to what extent the three selected features could be used to predict TOP2B binding in systems that were not used for training. Besides the two systems considered so far, we included TOP2B binding data from mouse activated B cells (Gene Expression Omnibus GSM2635606). We assessed model predictions by training on one system and testing prediction accuracy on the rest. Besides SVM and NB, this time we also applied Random Forests (RF) [36], which has been widely applied to genomic data with excellent results [37]. RF models performances were comparable to those of SVM and NB (S4 Table), and analysis of feature importances was consistent with the feature selection results (S7 Fig). The achieved predictions showed that cross-system applications of the classifiers did not decrease their performances, which acquired accurate predictions independently of the training system (Fig 4C and S8 Fig; Table 4). For example, when we applied NB models trained on MEF and mouse liver to activated B cells, we obtained accuracies of 98.3% and 97.9%, respectively, compared to an average of 97.6% using the model trained on activated B cells itself. These results support the idea that TOP2B association with DNase-seq, CTCF and RAD21 can be generalized across mouse systems.

## A general model of TOP2B binding based on DNase-seq, RAD21 and CTCF

Given the above observations, and to better capture the system specificities of TOP2B binding in a single classifier, we built a model trained with a combination of data from mouse liver, MEFs and activated B cells. We used this model to generate genome wide predictions of TOP2B binding in several systems, including those used for training and two additional systems that served as the validation sets (see next sections). To do so, we split the genome in 300

**Table 4. Performance of cross system predictions using models trained with DNase-seq, RAD21 and CTCF.** Errors are only shown for models trained and tested with the same system due to test data partition.

| System (train/test) | RF | NB | SVM |
| --- | --- | --- | --- |
| MEF | 95.03 ± 0.44 | 93.89 ± 0.23 | 95.22 ± 0.28 |
| MEF/Liver | 91.97 | 89.43 | 92.36 |
| MEF/aB cells | 98.31 | 98.26 | 98.38 |
| Liver | 94.90 ± 0.16 | 91.60 ± 0.51 | 92.57 ± 2.20 |
| Liver/MEF | 89.33 | 89.04 | 93.11 |
| Liver/aB cells | 96.38 | 97.87 | 98.21 |
| aB cells | 98.70 ± 0.22 | 97.63 ± 0.42 | 98.28 ± 0.31 |
| aB cells/MEF | 93.16 | 85.00 | 92.95 |
| aB cells/Liver | 89.08 | 85.49 | 81.19 |

bp windows, scored DNase-seq, CTCF and RAD21 signals and applied the TOP2B generalizing classifier within such windows, obtaining probability values through the whole genome for each system (S9 Fig).

The fact that only three features are sufficient to predict TOP2B binding genome wide opens the question of whether our model is truly informative, or similar performances could be achieved by simply considering the merged set of DNAse-seq, CTCF and RAD21 peaks. In order to test this, we focused on datasets in mouse liver, due to the excellent quality and reproducibility of the experimental TOP2B ChIPseq data [11]. We generated genome wide predictions and considered regions with a probability > 0.95 as predicted binding sites. Then, we compared our predictions and the set of merged DNase-CTCF-RAD21 peaks regarding their overlap with experimental TOP2B peaks (S10(A) Fig). We identified 68, 163 predicted sites, among which 56% coincided with ChIP-seq peaks. Regarding the 117, 714 DNase-CTCF-RAD21 detected peaks, this percentage decreased to 37%, suggesting that the merged-peak approach could lead to the selection of a large number of false positives. Indeed, while the specific potential false positives of our model (not identified by ChIP or by the merged set) clearly displayed TOP2B signal in heatmaps (S10(B) Fig, bottom), this was greatly reduced in the case of the specific DNase-CTCF-RAD21 set (S10(B) Fig, top). These results highlight the sensitivity of our model and shows that it clearly outperforms a simple DNase-CTCF-RAD21 peak merging by also capturing quantitative TOP2B binding information.

## TOP2B probability mirrors experimental binding intensity genome wide

Given the remarkable predictive power of our model, we used the genome wide TOP2B binding probabilities generated in the previous section (S9 Fig) as virtual tracks and represented them in a genome browser [26] together with experimental TOP2B ChIP-seq signals. Virtual tracks faithfully simulated ChIP-seq data at the gene level and accurately captured system binding differences in the training datasets (Fig 5 and S11 Fig). It is worth noting that, even though we modeled TOP2B binding using binary classifiers, the similarities between the experimental signals and our predicted probabilities are quite significant, showing Pearson's correlation coefficients comparable to those obtained across replicates (S5 Table). Furthermore, signal comparison with the three chromatin features used as predictors showed that the model was also able to apply learned associations of DNAse-seq, RAD21 and CTCF with TOP2B to precisely generate *bona fide* predictions (Fig 5 and S11 Fig). These results highlight the power of our model and demonstrate that the described approach can be used as a framework for the prediction of genome wide TOP2B signal potentially in any mammalian system for which sequencing data on DNAse I hypersensitivity, RAD21 and CTCF are available. Furthermore, this demonstrates that, besides global TOP2B binding, our model can be used to produce virtual probability tracks with which quantitative TOP2B accumulation can be analyzed at specific genomic loci.

## Validation in mouse thymocytes

To further test the predictive power of our approach, we generated genome wide predictions in mouse thymocytes and used TOP2B ChIP-seq data performed in our laboratory [60] to validate them. In order to test the specificity of our model, we evaluated the ChIP-seq signal generated by two different antibodies (Novus and Santa Cruz). Visual inspection of predictive and experimental tracks in a genome browser revealed that our predictions accurately mirror ChIP-seq signals (Fig 6A and S12 Fig). Furthermore, the predictions seemed to capture the learned associations of DNAse-seq, RAD21 and CTCF with TOP2B (S12 Fig), as observed for the training systems, and showed Pearson's correlation coefficients comparable to those

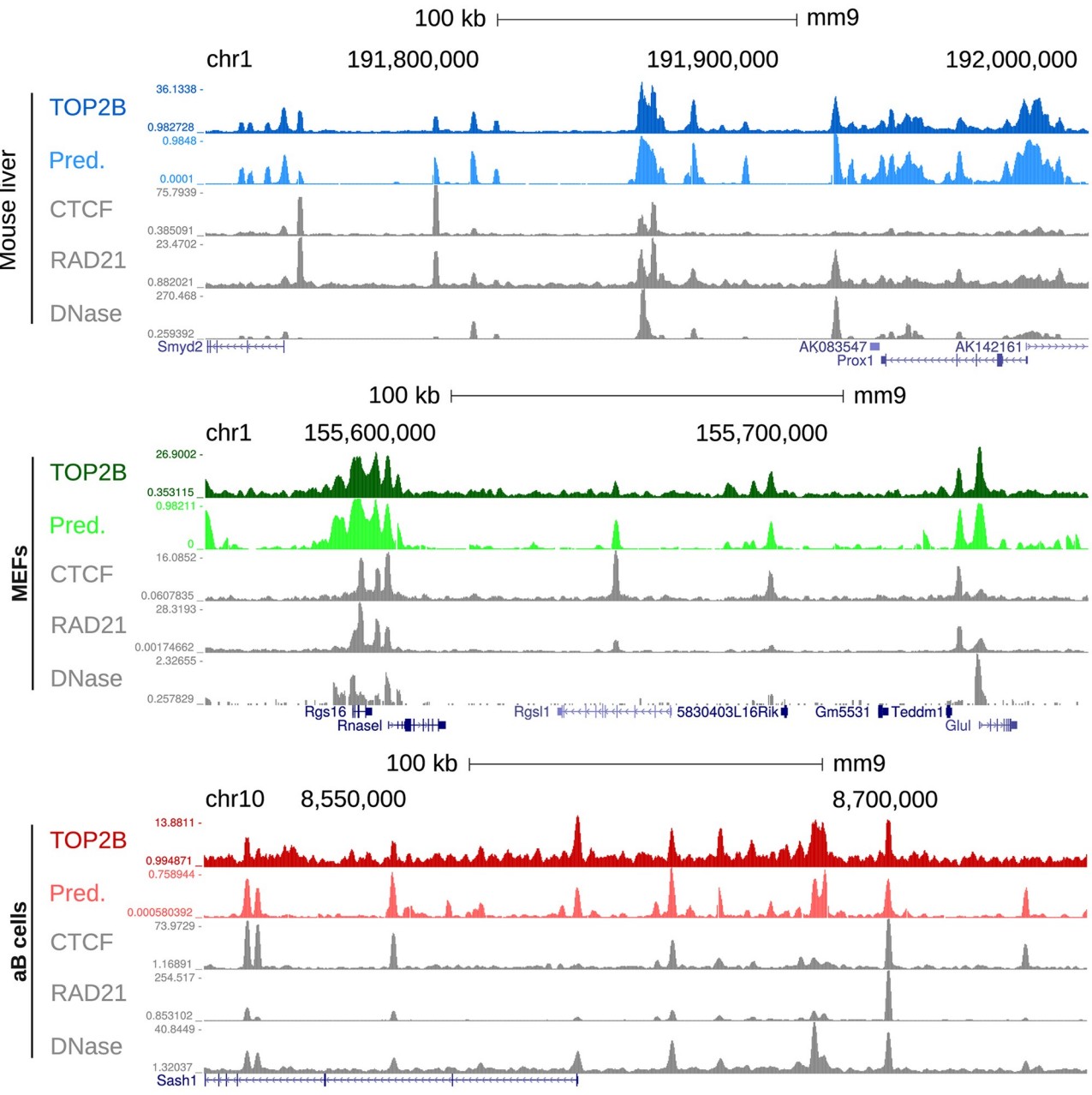

**Fig 5. TOP2B predictive and experimental tracks in mouse liver, MEFs and activated B cells (training systems).** From top to bottom, genome browser view of TOP2B ChIP-seq, TOP2B virtual track, CTCF ChIP-seq, RAD21 ChIP-seq and DNase-seq are displayed for each system.

obtained across replicates (S6 Table). Next, we called peaks using HOMER [25] and identified 2, 570 and 13, 297 TOP2B binding sites for Novus and Santa Cruz, respectively. Then, we applied the model and computed AUC values (Fig 6B). Although moderate predictive power was achieved for individual antibodies (AUC of 0.83 and 0.73 for Novus and Santa Cruz, respectively), when applying the model to common peaks of both experiments we obtained quite accurate predictions, with AUC of 0.96.

To further compare predictions with our experimental ChIP-seq data in mouse thymus, we performed peak overlap analysis. Again, we filtered our predictive track and kept only those

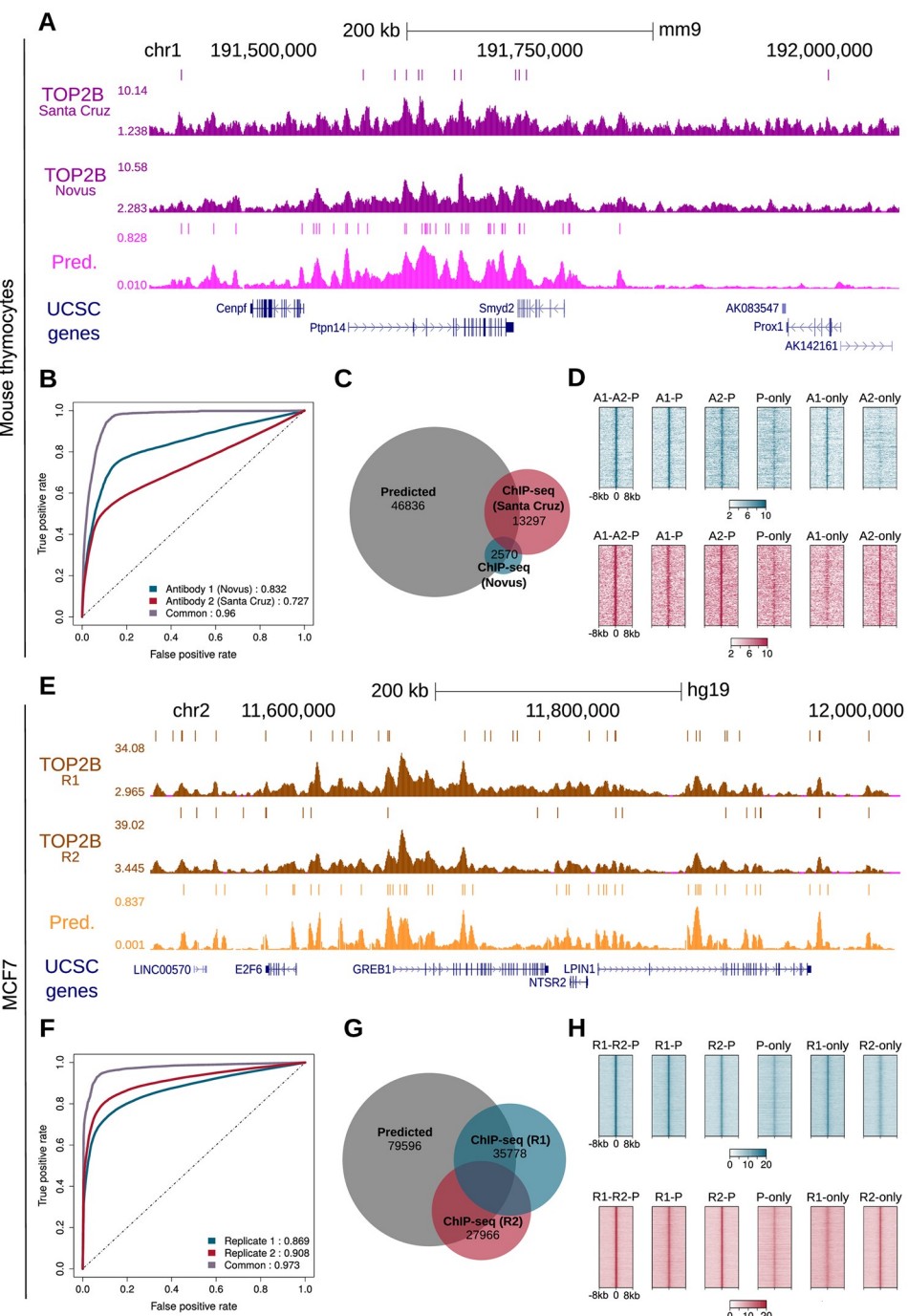

**Fig 6. Validation of TOP2B model in mouse thymocytes and human MCF7. A**. TOP2B predictive and ChIP-seq tracks of mouse thymocytes in a selected region of the mouse genome. Identified peaks are displayed above the signals. **B**. ROC curves and AUC values for the prediction of TOP2B peaks detected using Novus and Santa Cruz antibodies. **C**. Venn diagram showing the overlaps between predicted TOP2B peaks and the two sets of experimental peaks. **D**. Heatmap representations of TOP2B ChIP-seq reads enrichment within ± 8 kb of several set of peaks: predicted and confirmed by HOMER calls of both antibodies (A1-A2-P), predicted and confirmed only by Novus antibody (A1-P) or Santa Cruz (A2-P), only predicted (P-only), only detected by Novus (A1-only) and only detected by Santa Cruz (A2-only). For illustration purposes, the same number of randomly selected peaks is represented in all the heatmaps. **E**. TOP2B predictive and ChIP-seq tracks of MCF7 in a selected region of the human genome. Identified peaks are displayed above the signals. **F**. ROC curves and AUC values for the prediction of TOP2B peaks in two replicates performed in MCF7. **G**. Venn diagram showing the overlaps between predicted TOP2B peaks and the two sets of

experimental peaks. **F**. Heatmap representations of TOP2B ChIP-seq reads enrichment within ± 8 kb of several set of peaks: predicted and confirmed by HOMER calls of both replicates (R1-R2-P), predicted and confirmed only by the first (R1-P) or the second replicate (R2-P), only predicted (P-only), only detected by the first replicate (R1-only) and only detected by the second replicate (R2-only). As in **D**, the same number of randomly selected peaks is represented in all the heatmaps.

genomic windows with a probability of TOP2B binding above 0.95, which resulted in 46, 836 predicted peaks. Such peaks included 53.4% of experimental ChIP-seq peaks detected using Novus antibody, 30.7% of those detected using Santa Cruz and 72.6% of common peaks (Fig 6C). In addition, our model detected a set of 41, 766 TOP2B peaks that were not identified by HOMER in any of the two ChIP-seq experiments. In order to assess whether we were potentially detecting false positives, we examined ChIP-seq intensity centered at TOP2B peaks using heatmaps (Fig 6D). We displayed TOP2B at common and individually identified peaks, including (Fig 6D): predicted peaks confirmed by both experimental datasets (A1-A2-P), predicted and confirmed only by Novus antibody (A1-P) or Santa Cruz (A2-P), only predicted (P-only), only detected by Novus (A1-only) and only detected by Santa Cruz (A2-only). Interestingly, an enrichment of TOP2B signal was observed at all types of predicted peaks. As expected, A1-A2-P, A1-P and A2-P peaks showed a higher TOP2B enrichment, although peaks only predicted by our model still displayed an evident signal for both antibodies (Fig 6D). Visual inspection of such regions in a genome browser confirmed that, although less enriched than those confirmed by the antibodies, P-only peaks constitute TOP2B binding sites that were not identified by peak calling (S13 Fig). Furthermore, examination of A1-only and A2-only peaks showed that our predictive track displayed TOP2B probability at these regions, but they were not classified as predicted due to the threshold of 0.95 (S13 Fig). In summary, these observations indicate that our predictions are indeed highly sensitive to TOP2B binding even in a mouse system not used for generating the model.

## Validation in human cells

In order to test whether our TOP2B binding model can be generalized to other organisms, we generated predictions in the human MCF7 cell line and performed ChIP-seq of TOP2B to validate them. This time we performed two replicates using the same antibody (Sigma), whose specificity was validated for both human and mouse cells (S14 Fig), and allowed us to evaluate to what extent our predictions resemble an experimental replicate. Indeed, visual inspection in a genome browser revealed that TOP2B predictive track accurately mirrors both experimental signals (Fig 6E and S12 and S15 Figs), and Pearson's correlation coefficients were again comparable to those obtained across replicates (S7 Table). Then, we called HOMER and identified 35, 778 and 27, 966 TOP2B peaks for each replicate, with 10, 310 common peaks (Fig 6G). Our model obtained quite accurate predictions on both sets of peaks, with AUC of 0.87 and 0.91 (Fig 6F). When predicting on the set of common peaks, the model performed even better, with AUC of 0.97.

Again, we performed peak overlap analysis to further compare predictions with our experimental ChIP-seq data. We identified 79, 596 regions with a probability of TOP2B binding above 0.95, including 45.98% of experimental peaks from replicate 1, 56.75% of experimental peaks from replicate 2 and 74.45% of common peaks, showing that our predictions behave as a virtual experimental replicate (Fig 6G). Since our predictions revealed an additional set of 55, 311 peaks that were not detected by HOMER, we assessed whether these corresponded to potential false positives, as described above: predicted and confirmed by HOMER in both replicates (R1-R2-P), predicted and confirmed only by the first (R1-P) or the second replicate

(R2-P), only predicted (P-only), only detected by the first replicate (R1-only) and only detected by the second replicate (R2-only). We observed a similar pattern as in our previous analysis in mouse thymocytes, with predicted peaks confirmed by both antibodies showing the highest enrichment of TOP2B reads (Fig 6H). Again, we examined peaks only predicted by our model in a genome browser and confirmed that they corresponded to true binding sites (S15 Fig). On the other hand, R1-only and R2-only peaks were found to be enriched in our probability track, which indicates that they were not detected as predicted peaks because of the defined threshold of 0.95 (S15 Fig). These results confirm that the model is also accurate in predicting TOP2B binding in human cells.

## Prediction of TOP2-induced DSBs

Upon etoposide treatment, abortive TOP2 activity results in DSBs at CTCF-RAD21 sites, as measured by the END-seq method [12, 17]. Since TOP2-induced DSBs have important implications for cancer-linked translocations [12, 17, 18, 60, 61], we decided to test the capacity of our model to predict these breaks. Given that many of the datasets provided by Canela *et al* (2017) correspond to MEFs, we focused our analysis on this cell line and processed an END-seq sample treated with etoposide in addition to the already analyzed TOP2B ChIP-seq dataset (see S1 Table). A genome browser view of a chromatin loop near the *Lrp8* gene revealed an enrichment of END-seq and TOP2B ChIP-seq signals at the anchors, which was accurately mirrored by our predictive track (S16(A) Fig). We performed peak calling on the END-seq sample and identified 4, 363 DSBs. Then, we applied our model and computed AUC values, obtaining accurate predictions of END-seq peaks (AUC of 0.91), although still lower than those achieved on the ChIP-seq sample (AUC of 0.99) (S16(B) Fig). We next performed peak overlap analysis to further compare our genome wide predictions with the experimental DSBs. We used our previously generated TOP2B virtual track and filtered regions with a probability greater than 0.95. This resulted in 87, 205 predicted peaks, including 61.2% of experimental DSBs and 83.3% of ChIP-seq peaks (S16(C) Fig). Thus, our model achieves a reasonably high prediction accuracy when applied to TOP2-induced DSBs, but, as expected, is clearly better in predicting TOP2B binding. We argue that other factors beyond binding itself may be implicated in TOP2B activity and subsequent DSB formation.

## Conclusion

TOP2B is a crucial enzyme for DNA metabolism whose activity is required for a wide range of cellular processes including replication, transcription and genome organization. The few genome-wide maps of TOP2B that are currently available show high colocalization with several chromatin features, but no work has been devoted to comprehensively study predictive features that define TOP2B binding. By applying feature selection techniques, we show that TOP2B binding can be accurately predicted using only three high-throughput sequencing datasets, DNase-seq and ChIP-seq of RAD21 and CTCF, highlighting the importance of TOP2B in the regulation of genome architecture. Since the above datasets are largely available in public databases, our framework allows the scientific community to predict TOP2B binding in a large set of organisms, cell types, tissues and biological conditions. Finally, we believe that this approach and the generation of virtual probability tracks could be applied to other chromatin binding factors with a similar sequence-independent binding behavior to TOP2B.

## Supporting information

**S1 Fig. Genome browser view for TOP2B ChIP-seq signal and other relevant chromatin features in MEFs.** From top to bottom, TOP2B (grey), Pol2 (purple), RNA-seq (black),

DNase-seq (yellow), architectural factors (blue) and a selection of histone marks (green) are displayed.
(TIF)

**S2 Fig. Enrichment of CpG methylation around TOP2B binding sites in mouse liver and MEFs.** Average of whole genome bisulfite sequencing reads within ±1 kb of TOP2B binding center, random regions and GC corrected random regions are displayed.
(TIF)

**S3 Fig. Genomic distribution and signal enrichment of TOP2B training peaks. A.** In the left panel, peaks were grouped into five classes as described in Matthews and Waxman, 2018 (see Materials and methods): promoters, weak promoters, enhancers, weak enhancers and insulators. The right panel displays pie charts showing peaks distribution relative to mouse TSSs. Data correspond to mouse liver.**B.** Same as **(A)** for MEFs data. **C.** Heatmap representations of TOP2B ChIP-seq reads enrichment within ± 8 kb of training peaks.
(TIF)

**S4 Fig. Chromatin features predict TOP2B binding.** ROC curves and AUC values for Naive Bayes models trained on the indicated sets of features.
(TIF)

**S5 Fig. ROC curves and AUC values for Support Vector Machine and Naive Bayes models trained with selected features and the whole set of chromatin features.** Models trained with a selection of 3 features show similar or even better performance than models trained with the whole set of features.
(TIF)

**S6 Fig. ROC curves and AUC values for Support Vector Machine and Naive Bayes models trained in mouse liver using DNase, RAD21 and CTCF or DNase, RAD21 and STAG2.** Models trained with either set of chromatin features show similar performances.
(TIF)

**S7 Fig. Ranking of random forests feature importances.** Only high-throughput sequencing datasets were considered. The predictive ability of the chromatin features is consistent with the feature selection analysis using Scatter Search and Fast Correlation-Based Filter.
(TIF)

**S8 Fig. ROC curves and AUC values for Support Vector Machine and Random Forests models.** Models were trained on either MEF, liver or activated B cells and applied to the three systems. Only DNase-seq and ChIP-seq of RAD21 and CTCF were used for training.
(TIF)

**S9 Fig. Framework for the validation of TOP2B model in mouse and human.** First, the genomes of mouse and human were tiled into bins of 300 bp with sliding windows of 50 bp. Then, DNase-seq and ChIP-seq reads of CTCF and RAD21 were scored on those bins (see Materials and methods) and the TOP2B binding model trained on mouse liver, MEFs and activated B cells was applied. Bins having a TOP2B probability higher than 0.95 were classified as TOP2B binding regions. Validation was performed on thymocytes and MCF7 cells for mouse and human, respectively.
(TIF)

**S10 Fig. Comparison of TOP2B predictions with DNase-CTCF-RAD21 merged peaks and experimental TOP2B peaks in mouse liver. A**. Venn diagram showing the overlaps between

predicted TOP2B peaks (dark grey), DNase-CTCF-RAD21 peaks (light grey) and two replicates of experimental peaks (blue and red). **B**. Heatmap representations of TOP2B ChIP-seq reads enrichment within ± 8 kb of specific DNase-CTCF-RAD21 merged peaks (not predicted and not detected by TOP2B ChIP-seq) and specific predicted peaks (not included in the DNase-CTCF-RAD21 set and not detected by TOP2B ChIP-seq). The enrichment of TOP2B signal in the latter confirms the high sensitivity of our predictions. Peak calling was performed using HOMER. For illustration purposes, the same number of randomly selected peaks is represented in the two heatmaps.
(TIF)

**S11 Fig. TOP2B predictive tracks in mouse liver, MEFs and activated B cells (training systems).** From top to bottom, genome browser view of TOP2B ChIP-seq, TOP2B virtual track, CTCF ChIP-seq, RAD21 ChIP-seq and DNase-seq are displayed for each system. TOP2B predicted and ChIP-seq peaks are also shown.
(TIF)

**S12 Fig. TOP2B predictive track in mouse thymocytes and human MCF7 (validation systems).** From top to bottom, genome browser view for TOP2B ChIP-seq, TOP2B predicted track, CTCF ChIP-seq, RAD21 ChIP-seq and DNase-seq. TOP2B predicted and ChIP-seq peaks are also shown.
(TIF)

**S13 Fig. Comparison of regions only detected by our TOP2B predictor and regions only identified by peak calling on experimental ChIP-seq data (mouse thymocytes).** From top to bottom, genome browser view for TOP2B ChIP-seq using Novus and Santa Cruz antibodies, TOP2B predicted track, CTCF ChIP-seq, RAD21 ChIP-seq and DNase-seq. **A**. Highlighted in red are examples of true TOP2B binding sites only detected by our predictive approach. An example of one TOP2B binding site only detected by Santa Cruz antibody is highlighted in blue. Our TOP2B predictive signal displays an increased probability at such region, although lower than the threshold of 0.95.
(TIF)

**S14 Fig. Validation of anti-TOP2B antibody. A**. Western-blot of total MCF7 cell extract depleted (siTOP2B) or not (siC) for TOP2B with antibodies against TOP2B or $\alpha$-Tubulin as a loading control. **B.** As in (A) with extracts from wild type (WT) and TOP2B-knock out (KO) MEFs.
(TIF)

**S15 Fig. Comparison of regions only detected by our TOP2B predictor and regions only identified by peak calling on experimental ChIP-seq data (human MCF7 cell line).** From top to bottom, genome browser view for TOP2B ChIP-seq, TOP2B predicted track, CTCF ChIP-seq, RAD21 ChIP-seq and DNase-seq. Examples of true TOP2B binding sites only detected by our predictive approach are highlighted in red and examples of TOP2B binding sites only detected by either ChIP-seq sample replicate are highlighted in blue. Our TOP2B predictive signal displays an increased probability at such regions, although lower than the threshold of 0.95.
(TIF)

**S16 Fig. Comparison of TOP2B predictions with double strand breaks (DSBs) measured by END-seq upon etoposide treatment in MEFs. A.** Genome browser view illustrating TOP2B activity at loop anchors. From top to bottom, Hi-C contact enrichment, CTCF, RAD21 and TOP2B occupancy measured by ChIP-seq, DSBs enrichment measured by END-

seq and TOP2B predicted track. The left and right anchors of a loop near the *Lrp8* gene are highlighted by dark grey dashed lines. **B.** ROC curves and AUC values for the prediction of END-seq and ChIP-seq peaks using our generalizing model. **C.** Venn diagram showing the overlaps between predicted TOP2B peaks (dark grey), END-seq peaks (red) and ChIP-seq peaks (blue). Peak calling was performed using MACS2.
(TIF)

**S1 Table. Public and custom sequencing data used in this study.**
(XLSX)

**S2 Table. Ranking of features selected by scatter search and fast correlation base filter algorithms.**
(XLSX)

**S3 Table. Performance of Support Vector Machines and Naive Bayes classifiers trained either with DNAse, RAD21 and CTCF of DNase, RAD21 and STAG2 in mouse liver using GC-corrected background regions.**
(DOC)

**S4 Table. Confusion matrix corresponding to Ranfom Forests models trained with all chromatin features but DNA sequence and 3D DNA shape.** Percentage of correctly classified classes are shown.
(DOC)

**S5 Table. Pearson correlation between experimental TOP2B ChIP-seq replicates and predictions in mouse liver, MEFs and activated B cells.** ChIP-seq reads and probabilities were log2-transformed before comparison. Signal correlations were then computed at experimental ChIP-seq peaks indicated in the first column.
(DOC)

**S6 Table. Pearson correlation between experimental TOP2B ChIP-seq replicates and predictions in mouse thymocytes.** ChIP-seq reads and probabilities were log2-transformed before comparison. Signal correlations were then computed at experimental ChIP-seq peaks indicated in the first column.
(DOC)

**S7 Table. Pearson correlation between experimental TOP2B ChIP-seq replicates and predictions in MCF7.** ChIP-seq reads and probabilities were log2-transformed before comparison. Signal correlations were then computed at experimental ChIP-seq peaks indicated in the first column.
(DOC)

## Acknowledgments

We thank A. Herrero-Ruiz and C. Gómez-Marín for valuable comments. All the analyses were performed using custom scripts that were run on the High Perfomance Computing cluster provided by the Centro Informático Científico de Andalucía (CICA).

## Author Contributions

**Conceptualization:** Pedro Manuel Martínez-García, Felipe Cortés-Ledesma.

**Data curation:** Pedro Manuel Martínez-García.

**Formal analysis:** Pedro Manuel Martínez-García, Miguel García-Torres.

**Funding acquisition:** Federico Divina, Felipe Cortés-Ledesma.

**Investigation:** Pedro Manuel Martínez-García, Miguel García-Torres, Federico Divina, Francisco Gómez-Vela, Felipe Cortés-Ledesma.

**Methodology:** Pedro Manuel Martínez-García, Miguel García-Torres.

**Project administration:** Pedro Manuel Martínez-García, Felipe Cortés-Ledesma.

**Resources:** Federico Divina, Felipe Cortés-Ledesma.

**Software:** Pedro Manuel Martínez-García, Miguel García-Torres.

**Supervision:** Pedro Manuel Martínez-García, Felipe Cortés-Ledesma.

**Validation:** Pedro Manuel Martínez-García, José Terrón-Bautista, Irene Delgado-Sainz.

**Visualization:** Pedro Manuel Martínez-García, José Terrón-Bautista, Felipe Cortés-Ledesma.

**Writing – original draft:** Pedro Manuel Martínez-García, Miguel García-Torres, Felipe Cortés-Ledesma.

**Writing – review & editing:** Pedro Manuel Martínez-García, Felipe Cortés-Ledesma.

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
