## [Decision Letter · Decision Letter 0]

3 Jun 2020

Dear Dr. Cortés-Ledesma,

Thank you very much for submitting your manuscript "Genome-wide prediction of topoisomerase IIβ binding by architectural factors and chromatin accessibility" for consideration at PLOS Computational Biology.

As with all papers reviewed by the journal, your manuscript was reviewed by members of the editorial board and by several independent reviewers. In light of the reviews (below this email), we would like to invite the resubmission of a significantly-revised version that takes into account the reviewers' comments. Even though we expect you to address each comment, it seems to us to be essential that you particularly address concerns  about the distinction of the contribution of this work from ref 11 and other related literature, comparison to a simple baseline prediction (regions with CTCF, DNaseI and cohesin peaks), antibody validation, availability of the code (GitHub or similar) and data (please include the token within the main manuscript in addition to the cover letter), and clarification about feature selection strategy.  

We cannot make any decision about publication until we have seen the revised manuscript and your response to the reviewers' comments. Your revised manuscript is also likely to be sent to reviewers for further evaluation.

Sincerely,

Ferhat Ay, Ph.D

Associate Editor

PLOS Computational Biology

Alice McHardy

Deputy Editor

PLOS Computational Biology

Reviewer's Responses to Questions

**Comments to the Authors:**

Reviewer #1: The authors utlise a number of publically available chip seq data sets for TOP2B and also for CTCF, RAD21, STAG1, STAG2, and a number of histone modifications. They use a range of computing methods to compare the genomic locations of TOP2B peaks/enriched regions with the peaks for the other types of proteins (CTCF, RAD21 etc). They also look at chromatin accessibility (DNAase hypersentivity) and various DNA features including CpG islands.

They do find concordance between TOP2B peaks DNAase hypersenitive regions and CTCF and RAD21. However this is not surprising, as in ref 11 Uuskula-Reimand et al, they previously showed and reported in detail that half of CTCF and cohesin (contains RAD21) - part of abstract from Ref 11 copied below

"TOP2B associates with DNase I hypersensitivity sites, allele-specific transcription factor (TF) binding, and evolutionarily conserved TF binding sites on the mouse genome. Approximately half of all CTCF/cohesion-bound regions coincided with TOP2B binding. Base pair resolution ChIP-exo mapping of TOP2B, CTCF, and cohesin sites revealed a striking structural ordering of these proteins along the genome relative to the CTCF motif. These ordered TOP2B-CTCF-cohesin sites flank the boundaries of topologically associating domains (TADs) with TOP2B positioned externally and cohesin internally to the domain loop."

https://www.ncbi.nlm.nih.gov/pubmed/27582050

Although this reference is cited in the introduction of this study, this study fails to fully address how the findings in this manuscript relate to this previous closely related research. The authors should consider revising their Introduction and Discussion to reference more fully the closely related previous literature.

This study also reports the generation two new datasets one from mouse cells and one from human cells to utlise their machine learning programmes to determine if they can predict where the TOP2B peaks/enriched regions will be found. In the venn diagrams in Figure 5B and 5E approx half the predicted TOP2B sites (based on the location of CTCF, RAD21 and DNAse hypersentive sites) overlap with the experimentally determined TOP2B sites. This agrees with the 50% concordance published in ref 11. What might be the reasons for the concordance not being 100%?

This manuscript confirms what was was previously reported for murine cells, using different computer programmes.

Reviewer #2: In this manuscript Martinez-Garcia et al. develop a computational tool to predict the localization of TOPO2 beta genome wide based on input data sets from DNaseI, and Chip-Seq of CTCF and RAD21. The model can be used to predict both mouse and human TOPO2 beta localization. These results are based on the previously observed strong co-localization of TOPO2 beta with CTCF, cohesion and DNaseI accessible sites from several research groups in both mouse and human cells. The models show very good accuracy, with reported ~95% of TOPO2 beta peaks called using the trained sets with the models, and ~90% accuracy using the experimental data sets (MCF7, mouse thymocytes). Overall this is a well-done manuscript. However, given the strong co-localization of TOPO2 beta with the features used to develop the models (DNaseI, CTCF and cohesion), its utility is somewhat diminished. I recommended publication of this manuscript provided the following concerns are addressed.

Major Points

1.) It has been well known that TOPO2 beta co-localizes with DNaseI, CTCF and cohesion binding sites. It seems that much of the TOPO2 binding can be predicted summing the unique peak calls from the three chromatin elements (Fig 1C). What is the overlap frequency of TOPO2 beta peaks to the combined CTCF, DNaseI and Rad21 peaks (removing redundant peak calls – Bedtools Merge function)? Can this simple method achieve worse/similar/better accuracy that the computational methods? Can this analysis be added to Figure 5 (or as a supplement). A similar presentation can be used as in Figure 5 with the merged DNAseI, CTCF and Rad21 peak data set replacing the “Predicted” data set from the authors models. This could serve as good control as to the superiority of the authors models.

2.) It has been known for some time that regions of open chromatin are “sticky”. This has been seen when using IgG controls in ChIP-Seq experiments and when ChIPing proteins which do not associate with chromatin like GFP. This is a aprticulare problem if the washing is not stringent enough. I would like the authors to process an IgG control ChIP-Seq data set for one other their cell lines (at similar read density to the TOPO2 Beta data set), call peaks using similar parameters as used to call TOPO2 Beta peaks and determine the overlap with their TOPO2 Beta peaks and predicted peaks as seen in Figure 5 B, E. Can the analysis be included in the supplement?

3.) It is important to validate antibodies prior to using them for ChIP-Seq experiments. I don’t see that the authors have validated the TOPO2 Beta antibodies used for the ChjIP-Seq experiments. Can a extract of total cellular proteins be resolved on a 4-20% gradient gel and the antibodies used for ChIP-Seq be used to do a Western blot? Is one band observed at the expected molecular weight? Can this be added to the supplement?

4.) I don’t see anywhere in the manuscript that the code to run the model is available. I do see a GEO deposit number. Does this have the code to run the model? I am assuming it is not available to the reviewers as a reviewer link and password is not provided. The code needs to be made freely available. Detailed instructions should be provided on how to run the code, using CTCF, RAD21 and DNAseI inputs, to get the predicted TOPO2 Beta output.

Minor Points

1.) Can a key be added to 1B and 1D? as in figure 1C? It is difficult for the reader to go to the figure legend to determine what the groups are.

2.) Can the ChIP-Seq peak calls and the “predicted” peak cells be added for each track in Figure 6 A and B? Also, supplemental Fig S6, S7 and Fig 1A?

Reviewer #3: Summary:

Martinez-Garcia et al., built several machine learning models to predict the genome-wide binding of topoisomerase II beta (TOP2B) using sequence features, DNA shape, and chromatin properties obtained from publically available data. The authors showed that open chromatin and occupancy of chromatin architecture proteins are consistently the best predictors.Using only DNase I signal, CTCF, and RAD21 binding, the authors were able to accurately predict TOP2B binding. With experimental validations, the authors showed that the predictive power of these three features is conserved across cell types and between human and mouse, achieving performance similar to biological replicates. Furthermore the predicted probability of TOP2B binding highly resembles experimental data showing the potential of using such a model to achieve quantitative predictions.

We find the work by Martinez-Garcia et al. to be interesting and the manuscript well-written and should be of general interest to genome biologists and those with specific interests in DNA topoisomerases and DNA repair. This is a computational biology paper takes published observations that TOP2B correlates well with cohesin and DNA and formalizes them into a predictive model. Understanding TOP2B binding has important implications yet experiments are often limited by the availability of antibodies. Thus, the predictive model proposed in the manuscript is a valuable approach that could be applied by others. The authors evaluated different models and feature selection methods. Reasonable approaches appear to be taken both computationally and experimentally. For example, the authors carefully tested and interpreted the usage of GC-corrected background sets and used two different TOP2B antibodies for experimental validation.

Major comments:

The major concerns I have come from:

1) Apparent lack of availability of data (no reviewer token was given and so I could not check the if appropriate data and metadata was submitted. Small but crucial details such as the specific antibodies (e.g. no part number was provided) used were not included in the methods and there was no way to assess whether this, along with other crucial metadata was included in the GEO submission referred to.

2) Equally important would be to address the lack of code and intermediate files that would allow others, including reviewers, to replicate the findings. Given this paper uses existing and appropriate models to make useful predictions, it would be most important that other computational biologist can easily replicate them and make new predictions.

3) There seems to be no direct acknowledgement and exploration (aside from a citation) of recent papers that generated original data, and made computational models, to reveal and predict the location of TOP2 covalent complexes. Specifically the original END-seq method has been demonstrated to capture TOP2 covalent complexes (Canela et al. 2017 PMID: 28735753 and PMID: 31202577). In Canela et al. 2017 PMID: 28735753 they presented a linear regression model that could predict DNA breaks (TOP2cc’s) using only RAD21. It would be important for the authors to more directly relate their work to this study. Given this current study and the previous study used MEFs there would be specific opportunities to see how well the author’s current model predicts the END-seq (TOP2cc) data from Canela et al. 2017. At the very least some discussion is warranted giving both studies highlight the importance of cohesin in their predictions.

Detailed comments:

1) Method: For processing: “raw reads were merged for biological replicates”. For identification of binding sites: “when replicates were possible, peaks were called for individual replicates and overlapping peaks were kept”. It is thus not clear if biological replicates were merged prior to alignment or not.

2) Method: it is not exactly clear how the additional set of GC-matched controls was generated.

3) Method: the features used clearly violates the assumptions of NB. The authors did not clarify why they were still interested to try out the method.

4) Are the random controls showing similar genomic distribution (distance to TSS for example) as TOP2B peaks?

5) Line 233-234, 15 experiments used for model training is not clearly indicated in table S1. It is thus not clear which datasets were used for the classification shown in Table 1. This is then explained in lines 272-277. However, it will be more clear if it’s explained before getting into prediction results.

6) Interpretation: line 297, “These observations contrast with previous findings of TOP2B being involved in transcriptional regulation [47], and may indicate that other chromatin features, such as histone marks and DNase-seq, are capturing such association..” Would the authors expect difference performance if they used RNAP2 instead of RNA-seq? Could predictions for TOP2B at specific genomic regions (TSS/first exon) be a more appropriate question when RNA-seq is used? More details in the discussion how these results contrast previous results would be welcomed. For instance, the association of TOP2B occupancy with cohesin and DNAse I has been revealed in several studies as has its association with transcribed genes. Canela et al.’s END-seq work PMID: 28735753 clearly showed that without transcription a linear model with RAD21 binding could predict TOP2cc genome wide. To me it seems the results here from Martinez-Garcia et al are congruent with previous work. Discussing this would be relevant. Furthermore, the fact that transcription was relevant for TOP2 mediated DSBs is worth mentioning (using modified END-seq from Canela et al. PMID: 31202577 along with Gothe et al (PMID: 31202576)).

7) Analysis: Fig1D, GG signal peak seems to be biased on one side? Is this driven by a few regions?

8) Analysis: Fig6, it would be good to add a genome-wide correlation measure between prediction probability and real ChIP-seq data.

9) Analysis: The proposed model is trained using TOP2B peaks but the prediction is made on sliding windows across the genome. Could the authors train the model using whole-genome sliding windows or comment on the potential/necessity of doing so. Is it possible to adopt other models to be able to predict the quantitative signal (instead of classification)?

Reviewer #4: In the manuscript, the authors proposed a computational approach to predict TOP2B binding sites using chromatin accessibility and architectural proteins. The authors compared the performance of three classifiers: Naive Bayes, Support Vector Machine and Random Forests on a bunch of different features including: histone marks, Pol2 binding, architectural components, chromatin accessibility, gene expression, DNA shape, DNA sequence and CpG methylation. Next, they conducted feature selection and found that DNase I hypersensitivity, CTCF and cohesin binding are the most important features to predict TOP2B binding sites. Then, they trained a generalized model using these three features in one cell type and applied the model to a different cell type and validated the predictions with ChIP-seq of TOP2B, showing that the generalized model can predict TOP2B binding sites in new cell line and species.

TOP2B plays important roles in DNA metabolism and 3D organization of chromatin. But currently there are few TOP2B ChIP-seq datasets available. Based on the high accuracy of the predictions presented in the manuscript, this approach offers an attractive way to predict TOP2B binding in different cell types and tissues using the public available datasets. The manuscript is well written and is sound and accurate in general.

Comments

1. Why is Random Forests added after feature selection, not at the beginning of the comparing different features and feature selection? RF can also identify the most important features. Would the important features selected by RF be consistent with the ones selected by FCBF and SS?

2. In the “Feature selection algorithms” method section, the authors mention that FCBF uses SU as goodness function in line 124, and then mention CFS measure is used as goodness function in line 138. Is CFS used instead of SU in FCBF? Could the authors provide more details?

3. Could the authors add the comparison of models trained using all features vs three important features to validate that the prediction accuracy is not reduced much.

4. For feature selection, SS selects DNase-seq, RAD21 and STAG2 as important features in liver, while DNase-seq, RAD21 and CTCF in MEF. Could the authors compare the performance of the model trained using DNase-seq, RAD21 and CTCF with the model trained using DNase-seq, RAD21 and STAG2 in liver to support that the performance is similar?

5. The authors compared the predicted peaks with HOMER detected peaks. Did the authors compare this predictive model with other models that can predict TF binding sites (e.g. DRAF)?

6. Did the authors apply this approach to predict other TF binding sites, e.g. TOP2A? How is the performance of that?

7. The authors should provide more description about the machine learning framework. It is not clear what package if any was used. Code and scripts should be provided along with the training data. It would be nice to provide the folds of validation too.

**Have all data underlying the figures and results presented in the manuscript been provided?**

Reviewer #1: No: I could not view the new ChiP-seq data sets on GEO, it says they are private and not to be released until Nov 2022.

"Accession "GSE141528" is currently private and is scheduled to be released on Nov 16, 2022."

Are the machine learning scripts/programmes publically available?

If so where can they be accessed?

Reviewer #2: No: I don't see that the code needed for predicting the TOPO2 Beta occupancy is provided. I might be missing it.

Reviewer #3: No: see above comments

Reviewer #4: No: The authors don't provide any description of the code or software used to perform these experiments.

PLOS authors have the option to publish the peer review history of their article (what does this mean?). If published, this will include your full peer review and any attached files.

Reviewer #1: No

Reviewer #2: No

Reviewer #3: No

Reviewer #4: No
---

## [Decision Letter · Decision Letter 1]

10 Oct 2020

Dear Dr. Cortés-Ledesma,

Thank you very much for submitting your manuscript "Genome-wide prediction of topoisomerase IIβ binding by architectural factors and chromatin accessibility" for consideration at PLOS Computational Biology.

In light of the reviews (below this email), two reviewers agreed to the publication of this work. However, reviewer 2 has major concerns remaining and reviewer 4 has some minor issues. Therefore, we cannot accept the work in its current form but if all concerns are satisfactorily addressed in a revised version, we would be happy to consider this work again. Note that we will not be able to offer another round of revisions after this and will have to make an accept/reject decision. Your revised manuscript will be sent to at least one of the reviewers for further evaluation.

Sincerely,

Ferhat Ay, Ph.D

Associate Editor

PLOS Computational Biology

Alice McHardy

Deputy Editor

PLOS Computational Biology

Reviewer's Responses to Questions

**Comments to the Authors:**

Reviewer #1: OK now apart from a few small issues

1. how many million reads were each MCF7 replicate?

2. some typos - "split" not "splitted" in methods and on line 509

"extent" not "extend" on line 479

Reviewer #2: Reviewer 2

Many of the requested changes were not made to the satisfaction of the reviewer.

Major point 1 - The response to this point is very confusing. I think the authors did a good job of giving the reviewer what they wanted, but the result is very confusing. In supp Fig 10 the authors show that 100% of the R2 peaks overlap with the merged peaks, and ~70% with the predicted peaks. If that is the case why would not 70% of the rows in the Merged-only heat map analysis have the high enrichment scores as seen in the Predicted-only heat map analysis? Similarly for the R1? This heat map analysis does not make sense. The Merged-only heat map shows all very low enrichments, but 70% should be as high as those shown in the Predicted-only heat map.

Major point 2 - The requested analysis was not done. Maybe I was not completely clear. I wanted the authors to down load a IGG pulldown ChIP-Seq data set and use it in their peak calling algorithm and determine the overlap with their Top2 ChIP-Seq data sets and the predicted data sets. As is shown now it is very difficult to know what the authors did. I don't see any discussion of IGG controls in any of the text, methods. What was actually done?

Major point 3 - The validation is wholly inadequate. I requested a 4-20 gradient gel. What is shown is only a band which is about the correct MW for the target (Top2). I would like to see if the antibodies used in the study cross react with other cellular proteins.

Major point 4 - this has been addressed adequately.

Minor point 1 - this has been addressed adequately.

Minor point 2 - 6A- Novus is missing the peak calls, Fig 1A is missing all peak calls, S11, S12, S13, S15, S16 are missing peak calls. It would be best for the reader to see these peaks calls, to understand what is being called in each track and see visually the agreement/disagreement between the tracks.

Reviewer #3: I think the authors addressed my concerns as well as those of the other reviewers.

Reviewer #4: The authors have adequately addressed my comments and concerns. Their model has now been more rigorously tested and the manuscript is greatly improved.

There are some minor comments that need to be addressed:

1) Mislabeled Fig S4 as Fig S3 in multiple places: legend of Figure 3, line 363, line 375

2) Table 4 and S8 Fig: The cross-validation performance in Liver based on SVM (76.15 ± 9.76%) is lower than the cross-system performance (Liver/MEF 93.11% and Liver/aB cells 98.21%). Is the accuracy of the cross-validation performance in Liver based on SVM correct? In Figure S6, it looks that the cross-validation performance of SVM using RAD21, DNase and CTCF features in Liver is good (AUC 0.95).

3) In line 502-504, besides the Fig 5 examples, provide the Pearson's correlation of predicted TOP2B binding vs experimental signals like Table S5, S6 to validate that similarities are globally significant.

**Have all data underlying the figures and results presented in the manuscript been provided?**

Reviewer #1: Yes

Reviewer #2: Yes

Reviewer #3: Yes

Reviewer #4: Yes

PLOS authors have the option to publish the peer review history of their article (what does this mean?). If published, this will include your full peer review and any attached files.

Reviewer #1: No

Reviewer #2: No

Reviewer #3: No

Reviewer #4: No
---

## [Decision Letter · Decision Letter 2]

13 Nov 2020

Dear Dr. Cortés-Ledesma,

We are pleased to inform you that your manuscript 'Genome-wide prediction of topoisomerase IIβ binding by architectural factors and chromatin accessibility' has been provisionally accepted for publication in PLOS Computational Biology.

Best regards,

Ferhat Ay, Ph.D

Associate Editor

PLOS Computational Biology

Alice McHardy

Deputy Editor

PLOS Computational Biology

Reviewer's Responses to Questions

**Comments to the Authors:**

Reviewer #4: The authors have addressed the remaining issues.

**Have all data underlying the figures and results presented in the manuscript been provided?**

Reviewer #4: Yes

PLOS authors have the option to publish the peer review history of their article (what does this mean?). If published, this will include your full peer review and any attached files.

Reviewer #4: No

---

## [Editor Report · Acceptance letter]

12 Jan 2021

PCOMPBIOL-D-20-00455R2 

Genome-wide prediction of topoisomerase IIβ binding by architectural factors and chromatin accessibility

Dear Dr Cortés-Ledesma,

I am pleased to inform you that your manuscript has been formally accepted for publication in PLOS Computational Biology. Your manuscript is now with our production department and you will be notified of the publication date in due course.

With kind regards,

Jutka Oroszlan
